# Evolution of an assembly factor-based subunit contributed to a novel NDH-PSI supercomplex formation in chloroplasts

Yoshinobu Kato[1,3], Masaki Odahara [2] & Toshiharu Shikanai [1✉]

Chloroplast NADH dehydrogenase-like (NDH) complex is structurally related to mitochondrial Complex I and forms a supercomplex with two copies of Photosystem I (the NDH-PSI supercomplex) via linker proteins Lhca5 and Lhca6. The latter was acquired relatively recently in a common ancestor of angiosperms. Here we show that NDH-dependent Cyclic Electron Flow 5 (NDF5) is an NDH assembly factor in Arabidopsis. NDF5 initiates the assembly of NDH subunits (PnsB2 and PnsB3) and Lhca6, suggesting that they form a contact site with Lhca6. Our analysis of the *NDF5* ortholog in Physcomitrella and angiosperm genomes reveals the subunit PnsB2 to be newly acquired via tandem gene duplication of *NDF5* at some point in the evolution of angiosperms. Another Lhca6 contact subunit, PnsB3, has evolved from a protein unrelated to NDH. The structure of the largest photosynthetic electron transport chain complex has become more complicated by acquiring novel subunits and supercomplex formation with PSI.

[1] Department of Botany, Graduate School of Science, Kyoto University, Kyoto, Japan. [2] Biomacromolecules Research Team, RIKEN Center for Sustainable Resource Science, Saitama, Japan. [3] Present address: Graduate School of Agricultural and Life Sciences, The University of Tokyo, Tokyo, Japan. ✉email: shikanai@pmg.bot.kyoto-u.ac.jp

In respiration and photosynthesis, electron transport generates a proton motive force across the membrane, which drives $F_oF_1$-ATP synthase[1,2]. In eukaryotes, these reactions are performed in mitochondria and chloroplasts as the remnants of ancient prokaryotic energy production systems[3]. A series of redox reactions are accomplished by large multi-subunit membrane protein machineries such as complexes I–IV in mitochondria and photosystem (PS) I and PSII in chloroplasts.

Complex I (NADH dehydrogenase) and its counterpart in chloroplasts, the NADH dehydrogenase-like (NDH) complex, are the largest complexes in each electron transport pathway[4,5]. Complex I is the major entry point of electrons to the respiratory electron transport process in mitochondria: it accepts electrons through NADH oxidation via its N-module[6]. On the other hand, the photosynthetic NDH complex lacks N-modules and accepts electrons from reduced ferredoxin rather than NADH or NADPH depending on its unique structure that consists of the head of the Q module (subcomplex A, SubA) and electron donor-binding subcomplex (SubE)[7–9]. SubE includes a key subunit, NdhS (also known as CHLORORESPIRATORY REDUCTION 31, CRR31)[9], whose C-terminal segment has been suggested to contribute to ferredoxin binding via a fly-catching mechanism[8]. NdhV has been also suggested to contribute to ferredoxin binding[10]. P-module (membrane subcomplex, SubM) and SubA are conserved both in the respiratory and photosynthetic complexes and deliver electrons to plastoquinone, coupled with proton pumping[7,8] (Supplementary Fig. 1a). Subcomplex B (SubB) and lumenal subcomplex (SubL) are specific to the chloroplast NDH complex[11]. Electron flow from ferredoxin to plastoquinone mediates 'PSI cyclic electron flow', which produces an additional proton motive force without reducing $NADP^+$ and balances the ATP and NADPH production ratio[12,13]. In addition to energizing ATP synthesis, lumen acidification depending on the formation of proton motive force also triggers the thermal dissipation of excessively absorbed light energy from PSII[14] and down-regulates the activity of the cytochrome $b_6f$ (cyt $b_6f$) complex to prevent overloading of the electrons toward PSI[15]. In angiosperms, PSI cyclic electron flow consists of two pathways depending on PROTON GRADIENT REGULATION 5 (PGR5)/PGR5-like Photosynthetic Phenotype 1 (PGRL1) proteins and the NDH complex[16–18]. In Arabidopsis thaliana (Arabidopsis) mutants deficient in NDH activity, the size of the proton motive force is slightly lower than in wild-type (WT) plants[19]. Despite the contribution of the chloroplast NDH to proton motive force being smaller than PGR5/PGRL1, the contribution of the chloroplast NDH complex appears under low-light conditions[20,21], fluctuating light intensity[22], at low temperatures[23], and during induction of photosynthesis[24]. Double mutants that are defective in both of these cyclic electron flow pathways demonstrate a severely low-growth phenotype, indicating PSI cyclic electron flow to be required for efficient photosynthesis[16].

The chloroplast NDH complex further interacts with two copies of the PSI supercomplex consisting of a PSI core and four molecules of light-harvesting Complex I (PSI-LHCI) to form the NDH-PSI supercomplex[25], in the same way as respiratory Complex I forms a 'respirasome' with complexes III and IV in mitochondria[26]. In angiosperms, almost all of the NDH complex associates with two copies of PSI-LHCI via the linker proteins Lhca5 and Lhca6[25]. The NDH complex is sandwiched by PSI-LHCIs in the single-particle images observed by electron microscopy[27] (Supplementary Fig. 1b). The NDH-PSI supercomplex formation, especially via Lhca6, stabilizes the NDH complex[25]. In contrast, the contribution of Lhca5 to NDH stability was observed only in the lhca6 mutant. In the lhca5 lhca6 double mutant, the NDH complex was unassociated with any PSI-LHCI and was unstable[25].

Lhca6 was likely acquired in a common ancestor of angiosperms, whereas Lhca5 is also conserved in the moss Physcomitrella patens (Physcomitrella)[28]. The Lhca6-dependent supercomplex formation was acquired relatively recently in the evolutionary history of angiosperms. Lhca5 and Lhca6 are members of the Lhca family consisting of components of the PSI antennae complex, LHCIs[29]. Lhca6 originated from Lhca2, although the origin of Lhca5 is unclear. The stromal loop of Lhca6 was evolutionarily modified to switch its function from an antenna to a linker[25,30]. Lhca6 is in fact substituted for Lhca2 in a copy of PSI-LHCI interacting with the NDH complex, whereas Lhca5 is substituted for Lhca4 in another copy of PSI-LHCI[31] (Supplementary Fig. 1b). We also discovered that Lhca6 binds SubB of the NDH complex prior to the full assembly of the NDH complex and that SubB was the Lhca6 binding site[32]. However, the history of evolutionary modification of SubB in angiosperms to facilitate the novel supercomplex formation via Lhca6 remains unknown.

In this study, we analyzed the NDH-dependent cyclic electron flow 5 (NDF5) protein that is required for NDH activity[33]. The SubB subunits are severely destabilized in the Arabidopsis mutant defective in NDF5, indicating that the function of NDF5 is related to SubB[33]. NDF5 shows a similar amino acid sequence to one of the SubB subunits, PnsB2[33]. However, its exact molecular function has not been uncovered. Here, we show that NDF5 is not a subunit but an assembly factor of SubB. NDF5 formed an initial assembly intermediate of SubB with PnsB2 and PnsB3 subunits and Lhca6, indicating that the binding sites of Lhca6 on the NDH side are PnsB2 and PnsB3. We found evidence that the NDF5 gene was tandemly duplicated, and that PnsB2 has evolved from the duplicated NDF5 in a common ancestor of angiosperms. The PnsB3 subunit also likely evolved from an NDH-unrelated protein such that both subunits provide a binding site for Lhca6. Consequently, acquisition of these subunits contributed to the establishment of a novel supercomplex structure that requires Lhca6. The molecular size of the largest protein complex in photosynthetic electron transport pathway is still growing.

## Results

**NDF5 is an assembly factor of SubB.** NDF5 is required for NDH activity in vivo and has a sequence similarity with PnsB2[33], but its exact molecular function has not yet been clarified. Since the function of NDF5 seemed to be related to SubB[33], we analyzed the NDF5 accumulation in mutants defective in each SubB subunit (Fig. 1). SubB subunits (PnsB1–PnsB5 and PnsL3) depend on each other for stability, and the lack of one subunit destabilized all other SubB subunits[25] (Fig. 1). However, the accumulation level of NDF5 was severely reduced only in the pnsb2 and pnsb3 mutants, and the same or even higher levels of NDF5 were detected in other mutants (Fig. 1). The stability of NDF5 depends on PnsB2 and PnsB3, and the function of NDF5 may be related to these subunits.

NDF5 is unlikely to be a subunit of the NDH complex, because it was not detected in the proteomic analysis of the NDH-PSI supercomplex excised from the Blue Native (BN) gel[25]. To test this idea, we solubilized the NDH-PSI supercomplex and other protein complexes from the thylakoid membrane and separated them using sucrose density gradient (SDG) ultracentrifugation. Protein complexes are separated more gently in SDG ultracentrifugation than in BN-PAGE[32]. LHCII monomer, LHCII trimer, PSII monomer, PSI-LHCI, and the NDH-PSI supercomplex were separated as distinct green bands in SDG (Supplementary Fig. 2a). Using specific antibodies, NDF5 and PnsB2 proteins were probed in each SDG fraction using two-dimensional SDS-PAGE and immunoblotting. The peak of PnsB2 was detected in the fractions

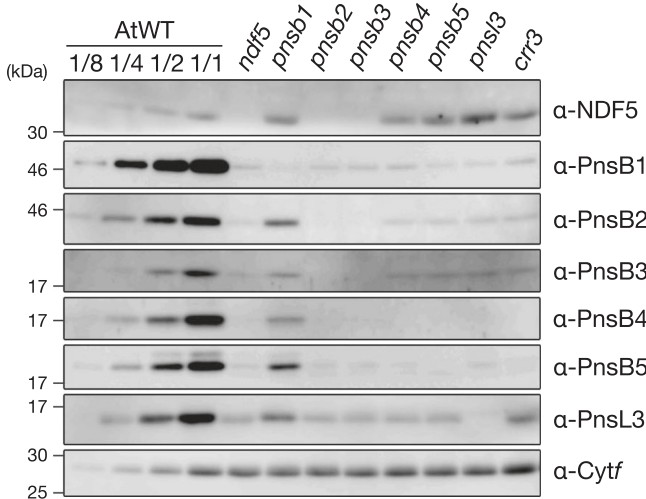

**Fig. 1 The accumulation of NDF5 depended on PnsB2 and PnsB3.**
Chloroplast membrane proteins isolated from WT and SubB mutants including *ndf5* were analyzed by immunoblotting. Sample loading was based on chlorophyll content, along with a dilution series of Arabidopsis wild-type (AtWT). Cyt*f* was detected as a loading control. Each experiment was performed at least twice, and independent results are shown in Supplementary Fig. 11a.

corresponding to the NDH-PSI supercomplex (fractions 23−25) in the WT Arabidopsis (Fig. 2a). However, the NDF5 peak was not a coincident with that of PnsB2, and was detected in fractions with less mobility (fractions 20 and 21) in the WT SDG (Fig. 2a). This result also suggests that NDF5 is not a subunit of the NDH complex.

NDF5 formed a peak in SDG between PSI-LHCI and the NDH-PSI supercomplex (Fig. 2a), suggesting that NDF5 forms a putative large protein complex (hereinafter the NDF5 complex). In the mutant defective in NDF5, SubB was more severely destabilized than other parts of the NDH complex[33] (Fig. 1). NDF5 may be involved in the assembly of SubB by interacting with SubB subunits. To assess this possibility, we detected the NDF5 complex in mutants defective in SubB subunits (PnsB1–PnsB5 and PnsL3). Whereas the NDF5 complex was formed in the *pnsb1*, *pnsb4*, *pnsb5*, and *pnsl3* mutants as in the WT, no fractions contained NDF5 in the *pnsb2* and *pnsb3* mutants (Fig. 2b), consistent with the accumulation level of NDF5 in these mutants (Fig. 1). The accumulation of NDF5 and the formation of the NDF5 complex depended on PnsB2 and PnsB3, suggesting that NDF5 interacts with these subunits.

If the above hypothesis were correct, PnsB2 and PnsB3 would be detected in the same SDG fractions as NDF5. In WT plants, the PnsB2 signal detected at the position of the NDH-PSI supercomplex was so strong that the tail of the signal masked the position of the NDF5 complex. Consequently, we could not distinguish the signal for the NDF5 complex from the mature NDH-PSI supercomplex using anti-PnsB2 antibody (Fig. 2a, α-PnsB2 long-exposure). Our strategy was then to detect PnsB2 and PnsB3 in the SDG fractions of the *pnsb1*, *pnsb4*, *pnsb5*, and *pnsl3* mutants, in which the NDF5 complex is formed but the NDH-PSI supercomplex is absent. Although the majority of PnsB2 and PnsB3 were detected in low-molecular-weight fractions (1−10) because of their destabilization due to the lack of a mature NDH-PSI supercomplex, the peaks of PnsB2 and PnsB3 were also detected at the position of the NDF5 complex (fractions 20 and 21) in the *pnsb4*, *pnsb5*, and *pnsl3* mutants (Fig. 2c). In the *pnsb1* mutant, most of the PnsB2 and PnsB3 proteins were detected in fractions 20 and 21 (Fig. 2c). This is because SubB is partially

stabilized even in the absence of PnsB1, which is incorporated at the final step in the SubB assembly[32]. In summary, in Fig. 2b, c, NDF5, PnsB2, and PnsB3 showed almost the same peak in the SDG fractions of the SubB mutants, except for *pnsb2* and *pnsb3*. This result is consistent with the proposition that NDF5 forms a protein complex with PnsB2 and PnsB3.

As described above, accumulation of NDF5 was drastically impaired in the mutants lacking PnsB2 or PnsB3 (Fig. 1), suggesting PnsB2 and PnsB3 to be essential for stabilizing NDF5. Since the SubB subunits depend on each other for stability, PnsB2 and PnsB3 were also severely decreased in the mutants defective in other SubB subunits[25,34–36] (Fig. 1). Nevertheless, the accumulated level of NDF5 was almost unaffected in the *pnsb1*, *pnsb4*, *pnsb5*, and *pnsl3* mutants (Fig. 1). NDF5 was stabilized by PnsB2 and PnsB3 in these mutants (Figs. 1 and 2b, c), i.e., it was stabilized by immature SubB. Taking the above results together, we propose that NDF5 is an assembly factor of SubB and forms an assembly intermediate that includes at least PnsB2 and PnsB3. We also found the NDF5 protein complex to be smaller in the mutants that were defective in Lhca6, which is a linker protein that mediates supercomplex formation between NDH and PSI-LHCI. This suggests that Lhca6 is also included in the NDF5 complex (Fig. 2b), and is consistent with the SubB assembly model, in which Lhca6 is incorporated into the SubB assembly intermediates prior to the completion of the full NDH complex assembly[32].

When does NDF5 come into play during the SubB assembly process? To answer this question, we analyzed the SubB assembly intermediate in the SDG fractions of the *ndf5* mutant. We were unable to analyze PnsB5 in the SDG fractions, because there was a non-specific signal which migrated to the same position as PnsB5 in SDS-PAGE. PnsB2–PnsB4 and PnsL3 were present only in low-molecular-weight fractions (fractions 1−13); none of their peaks were detected in high-molecular-weight fractions (Supplementary Fig. 3a). A weak PnsB1 signal was detected from fractions 24−25 (Supplementary Fig. 3a), which correspond to the NDH-PSI supercomplex (Fig. 2a). This peak depends on the leaky assembly of the NDH-PSI supercomplex independently of NDF5, and was only detected by the antibody against PnsB1, probably because of its higher titer. A similar situation was reported for the mutant defective in CRR3, which is another SubB assembly factor[32]. These assembly factors are required for efficient operation of the SubB assembly process, and actually the NDH activity in vivo was no longer detectable without them in the analysis of chlorophyll fluorescence[33,37]. Except for the minor peak at fractions 24−25, PnsB1 was also detected only in low-molecular-weight fractions (Supplementary Fig. 3a). Moreover, the peaks of PnsB1–PnsB4 and PnsL3 were not a coincident with each other, suggesting that they did not form any assembly intermediates in the *ndf5* mutant. It appears that NDF5 is required for efficiently initiating the early stage of SubB assembly. On the other hand, the peaks of PnsB2 and PnsB3 were detected in high-molecular-weight fractions (fractions 20 and 21) in the mutant lacking CRR3 (Supplementary Fig. 3b). NDF5 was also detected in those fractions (Supplementary Fig. 3b), and its accumulation level was not affected in the *crr3* mutant (Fig. 1). These results also indicate that some SubB assembly processes can proceed without CRR3 and that NDF5 works in earlier steps of SubB assembly than CRR3.

The accumulation levels of CRR3 and another SubB assembly factor, PsbQ-Like Protein 3 (PQL3) depend on the leaf-development stages in Arabidopsis[32,38]. It is abundant in immature leaves, in which the NDH complex is actively synthesized. We also analyzed the accumulation of NDF5 during the course of leaf development. Leaves were allocated to four groups according to leaf age (Fig. 3a). Chloroplast membrane

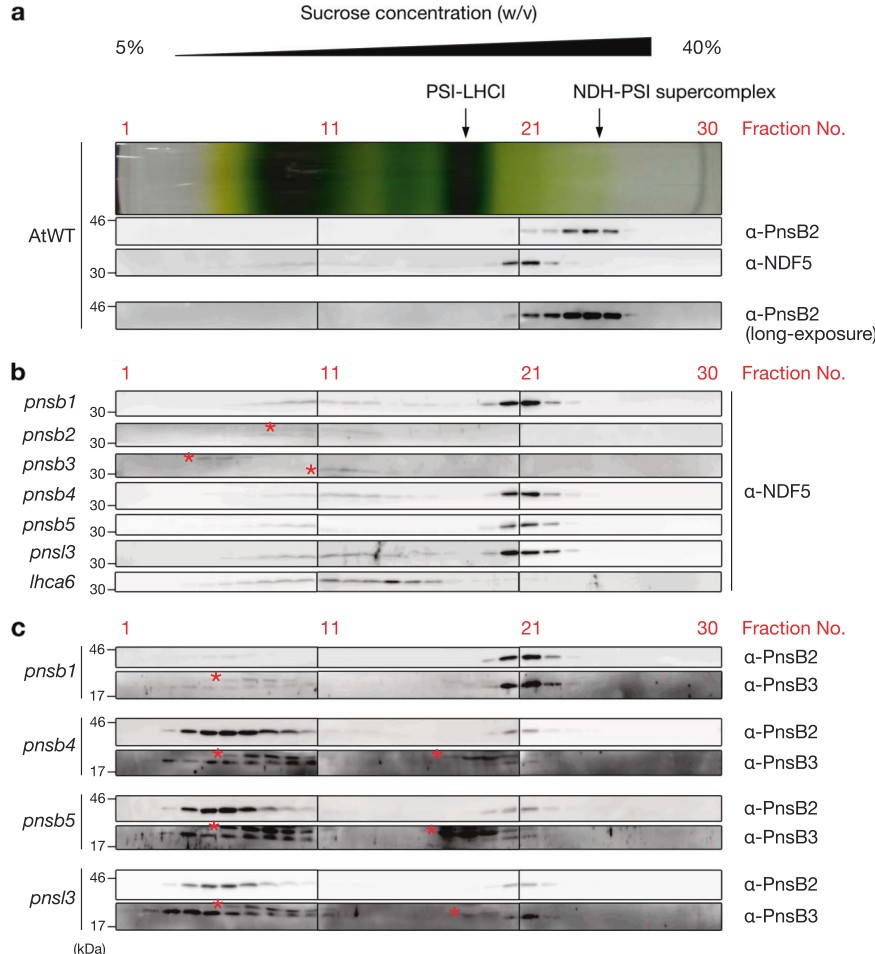

**Fig. 2 NDF5 formed a putative protein complex depending on PnsB2, PnsB3, and Lhca6.** Protein complexes of the chloroplast membrane isolated from Arabidopsis wild-type (AtWT) (**a**) and SubB and *lhca6* mutants ((**b**) and (**c**)) were separated using SDG ultracentrifugation. The top of the centrifugation tube is at the left. After centrifugation, the SDG was divided into 30 fractions from top to bottom, and the fractions subjected to immunoblot analysis. The centrifugation tube of the AtWT is shown as a representative pattern on the top in (**a**). The centrifugation tube and blotting patterns were fitted according to Coomassie Brilliant Blue-stained gels (Supplementary Fig. 2b). The positions of the NDH-PSI supercomplex and PSI-LHCI are indicated with black arrows. Asterisks indicate non-specific signals. Each experiment was performed at least twice.

proteins were loaded onto the SDS-PAGE based on chlorophyll amount. Total protein compositions and the level of Cyt*f* were constant between the leaf stages, indicating that serious senescence did not occur, even at stage 4 (Fig. 3b, c). PnsB2 and NdhH (a SubA subunit) were also almost equally accumulated in the course of leaf development. On the other hand, the level of NDF5 was gradually decreased as the leaves got older (Fig. 3c, stages 3 and 4), as observed for CRR3. This observation is consistent with the notion that NDF5 is not a subunit but an assembly factor of SubB.

**Bryophytes encode an orthologous gene for PnsB2 or NDF5.** Since PnsB2 and NDF5 resemble each other[33], both may have originated from the same ancestral protein. If this were the case, was the ancestral protein an assembly factor or a subunit? To address this question, we searched for PnsB2 and NDF5 orthologs in the liverwort *Marchantia polymorpha* (Marchantia) and in the moss *Physcomitrella patens* (Physcomitrella). The amino acid sequences of Arabidopsis PnsB2 and NDF5 were used as queries in Phytozome V12 Blastp (https://phytozome.jgi.doe.gov/pz/portal.html). Mapoly0135s0022 in Marchantia and Pp3c25_5270V3 in Physcomitrella were the best hit against both PnsB2

and NDF5 (Fig. 4a and Supplementary Tables 1 and 2). No other proteins with an E-value of less than 1.00E−10 were found against PnsB2 nor NDF5 (Supplementary Tables 1–3), suggesting that Marchantia and Physcomitrella possess either PnsB2 or NDF5. Since the function of these proteins was unclear, we call these proteins in bryophytes Protein X and its gene *Gene X*. In the Arabidopsis proteome, NDF5 was the best hit against Protein X of Marchantia and Physcomitrella as queries (Fig. 4a and Supplementary Table 3). Protein X might be an NDF5 ortholog; however, in the phylogenetic tree, Protein X branches from the root of the PnsB2 and NDF5 clades of angiosperms (Fig. 4b).

**Protein X of Physcomitrella functions as NDF5 rather than PnsB2 in Arabidopsis.** To elucidate the function of Protein X, we knocked out *Gene X* in Physcomitrella using a gene-targeting technique. Its coding region was replaced by the *Aph4* cassette, which confers hygromycin resistance on transformants (Supplementary Fig. 4a). *Gene X* is a single-copy gene in the Physcomitrella genome. Genomic polymerase chain reaction (PCR) confirmed the successful *Gene X* knockout (Supplementary Fig. 4b). The expression of *Gene X* was below the detection limit in RT-PCR (Fig. 5a). The knockout lines can grow as normally as the WT

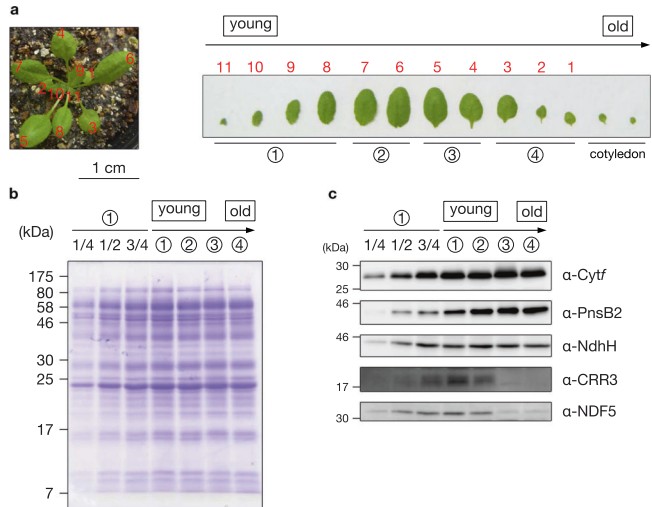

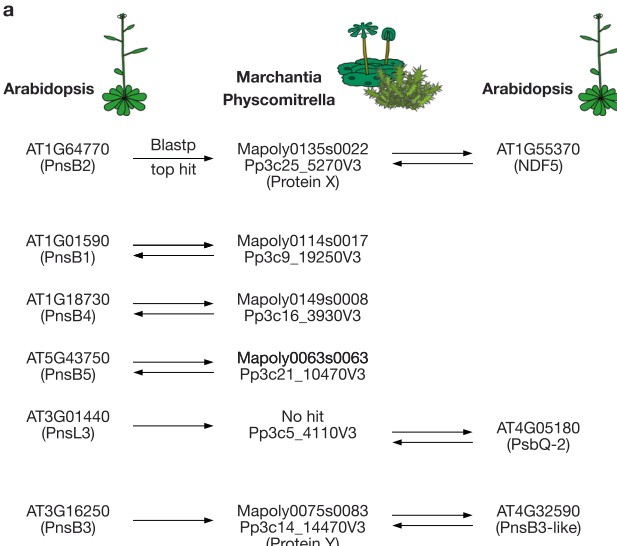

**Fig. 3 NDF5 was accumulated mainly in young leaves. a** Arabidopsis wild-type seedling grown for 28 days after germination (left). Detached leaves were allocated to four groups according to leaf stage, with the exception of cotyledons (right). **b, c** The chloroplast membrane proteins of each leaf stage were subjected to SDS-PAGE, and the gel was stained with Coomassie Brilliant Blue (**b**) or analyzed by immunoblotting (**c**). Sample loading was based on chlorophyll content, along with a dilution series of the stage 1 sample. Each experiment was performed at least twice, and independent results are shown in Supplementary Fig. 11b, c.

(Supplementary Fig. 4e). NDH activity can be monitored as a post-illumination transient increase in chlorophyll fluorescence[39]. Although this measurement is not quantitative, it is suitable for monitoring the absence of the NDH complex in various plant species including Arabidopsis and Physcomitrella[40,41]. In the WT, chlorophyll fluorescence increases transiently after turning off the actinic light (Fig. 5b). This fluorescence change was not observed in mutants defective in NDH activity[41] and this was similarly the case in the *Gene X* knockout mutant (Fig. 5b). Consistent with this result, the level of PnsB1 was severely decreased in the *Gene X* knockout mutant (Fig. 5c), indicating that *Gene X* in Physcomitrella encodes an NDH-related protein.

We could not obtain any specific antibodies against Protein X and could not biochemically analyze the function of Protein X in Physcomitrella. Instead, we tested whether Protein X could complement the functions of PnsB2 or NDF5 in the corresponding Arabidopsis mutants. *Gene X* was expressed under the control of the *35S* promoter (*35Sp*) in both mutants, and NDH activity was analyzed. Transient chlorophyll fluorescence changes were observed in WT but not in *pnsb2* and *ndf5* plants (Fig. 6a). None of the 39 plants independently transformed by the *35Sp::Gene X* construct in the *pnsb2* background showed a transient increase in chlorophyll fluorescence. On the other hand, NDH activity was detected in 12 out of 29 plants transformed by the *35Sp::Gene X* construct in the *ndf5* background. The results of representative transformant lines are shown in Fig. 6a and Supplementary Fig. 5a, and these lines were selected for further biochemical analysis. Because we faced a problem of transcriptional silencing of *Gene X* in T2 generation, we analyzed independent T1 lines. Low expression levels of *Gene X* may have interrupted the complementation of the *pnsb2* mutant. However, lines #23 and #24 in the *pnsb2* background accumulated higher levels of *Gene X* transcript than in lines #19 and #23 in the *ndf5* background, as indicated by RT-qPCR (Fig. 6b and Supplementary Fig. 5b). We also tested the recovery of the NDH-PSI supercomplex formation in the transgenic plants by BN-PAGE. The NDH-PSI super-complex was separated at the top of the BN gel but was absent in

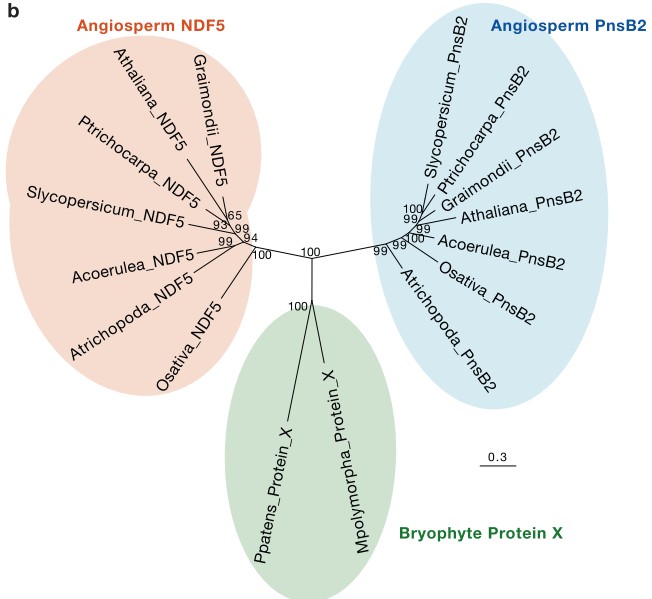

**Fig. 4 Summary of top hits in the reciprocal Blastp search between Arabidopsis and bryophytes. a** SubB subunits, NDF5, PsbQ-2, and PnsB3-like in Arabidopsis were searched in bryophytes, Marchantia and Physcomitrella. Each best hit was searched again in Arabidopsis. The black arrows indicate each best hit from queries. The E-value threshold was set at 1.00E⁻¹⁰. The results are shown in more detail in Supplementary Tables 1–3. **b** A phylogenetic tree of Protein X in bryophytes and NDF5 and PnsB2 in some angiosperms constructed based on Bayesian inference. Posterior probabilities for Bayesian inference are indicated. Branch length represents the estimated rate of amino acid substitution. Atrichopoda, *Amborella trichopoda*; Acoerulea, *Aquilegia coerulea*; Slycopersicum, *Solanum lycopersicum*; Graimondii, *Gossypium raimondii*; Ptrichocarpa, *Populus trichocarpa*; Athaliana, *Arabidopsis thaliana*; Osativa, *Oryza sativa*; Ppatens, *Physcomitrella patens*; Mpolymorpha, *Marchantia polymorpha*.

the *pnsb2* and *ndf5* mutants (Fig. 6c and Supplementary Fig. 5c). Consistent with the results of the activity analysis, the NDH-PSI supercomplex was formed in the *35Sp::Gene X/ndf5* plants but not in the *35Sp::Gene X/pnsb2* plants (Fig. 6c and Supplementary Fig. 5c). Finally, we analyzed the accumulated level of PnsB1 to quantitatively evaluate the complementation in the *35Sp::Gene X/ndf5* plants. The level of PnsB1 in these plants was about half of

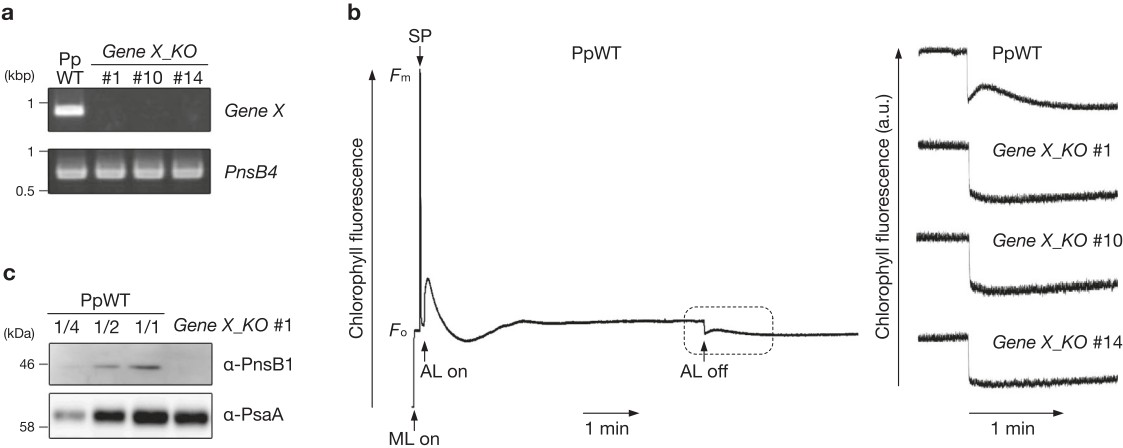

**Fig. 5 The NDH complex was disrupted in the *Gene X* knockout mutant in Physcomitrella. a** RT-PCR analysis of *Gene X* and *PnsB4* transcripts. These transcripts were amplified using cDNA from Physcomitrella wild-type (PpWT) and *Gene X* knockout mutants (*Gene X_KO*) mutants. *PnsB4* was detected as a control. **b** Analysis of transient increases in chlorophyll fluorescent after turning off actinic light (AL). A typical trace of chlorophyll fluorescence in the PpWT is shown at left. Plants were exposed to AL for 5 min, and the subsequent transient increase in chlorophyll fluorescence (boxed area) was monitored in the dark. Fluorescence levels were normalized against the maximum chlorophyll fluorescence ($F_m$) levels. Boxed area is magnified at right. $F_o$, minimal chlorophyll fluorescence; ML, measuring light; SP, saturation pulse of white light. **c** Immunoblot analysis of membrane proteins from PpWT and *Gene X_KO* #1. Samples were loaded on the basis of chlorophyll content, and PnsB1 and PsaA (a loading control) were detected. Each experiment was performed at least twice, and results in independent transgenic plants are also shown in Supplementary Fig. 11d.

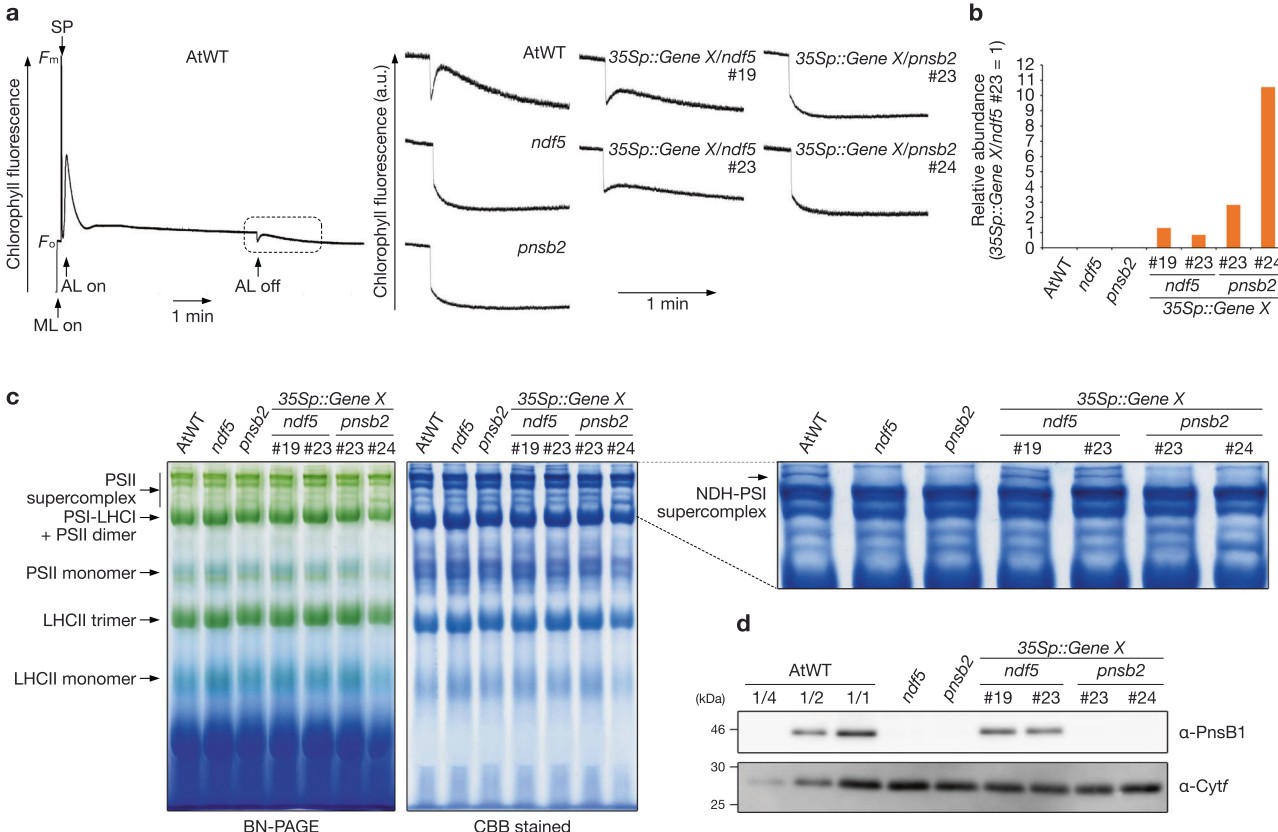

**Fig. 6 *Gene X* in Physcomitrella can complement the Arabidopsis *ndf5* mutant. a** Transient increases in chlorophyll fluorescent were monitored after turning off AL in Arabidopsis wild-type (AtWT), *ndf5*, *pnsb2*, *35Sp::Gene X/ndf5*, and *35Sp::Gene X/pnsb2*, as in Fig. 5b. A typical trace of chlorophyll fluorescence in the AtWT and transient increases in chlorophyll fluorescence (boxed area) are shown at left and right, respectively. **b** The expression level of *Gene X* in AtWT, mutants, and transgenic plants, as analyzed by RT-qPCR. **c** Separation of photosynthetic protein complexes by BN-PAGE. The gel was stained with Coomassie Brilliant Blue (right panel). Each band was identified according to previous work[71] and is indicated by black arrows at left. The top part of the BN gel is magnified at right. **d** Immunoblot analysis of chloroplast membrane proteins isolated from AtWT, mutants, and transgenic plants. Sample loading was based on chlorophyll content. Cyt*f* was detected as a loading control. This is a result of a single experiment using a T1 plant. The results using independent T1 plants are also shown in Supplementary Fig. 5.

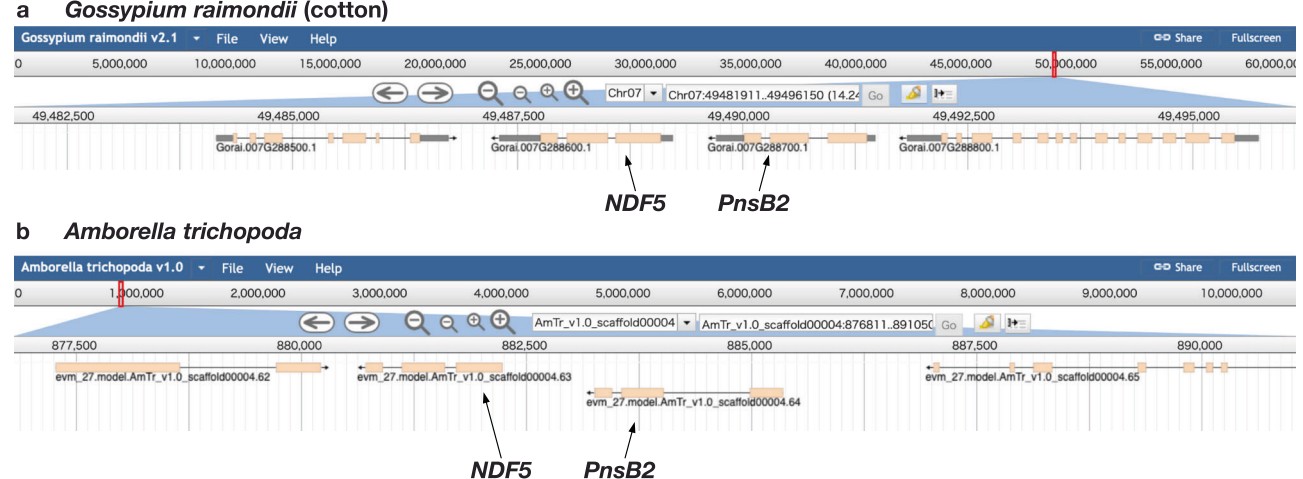

**Fig. 7 NDF5 and PnsB2 were tandemly arranged in *Gossypium raimondii* and *Amborella trichopoda* genomes.** Genome browser (Jbrowse in Phytozome V12; https://phytozome.jgi.doe.gov/jbrowse/) snapshots show the location of *NDF5* and *PnsB2* genes in (**a**) *Gossypium raimondii* and (**b**) *Amborella trichopoda* genomes. Other snapshots are shown in Supplementary Fig. 6.

that in WT plants, whereas it was below the detection limit in the *ndf5*, *pnsb2*, and *35Sp::Gene X/pnsb2* plants (Fig. 6d and Supplementary Fig. 5d). *Gene X* can partially rescue the *ndf5* mutant, suggesting that Protein X in Physcomitrella has a similar function to NDF5 and can act as an assembly factor. Protein X is likely NDF5 rather than PnsB2.

**The ancestral gene of *NDF5* and *PnsB2* tandemly duplicated during the evolution of angiosperms.** In Physcomitrella, Protein X (PpNDF5) probably functions as an assembly factor for SubB of the NDH complex. It is possible that the PnsB2 subunit originated from NDF5 via a gene duplication during the evolution of angiosperms. Indeed, we discovered that *NDF5* and *PnsB2* genes were tandemly arranged in some genomes of angiosperms (Fig. 7 and Supplementary Fig. 6). For example, in cotton (*Gossypium raimondii*), *NDF5* and *PnsB2* were tandemly encoded in *Gorai.007G288600* and *Gorai.007G288700*, respectively (Fig. 7a). A tandem arrangement of *NDF5* and *PnsB2* was observed in some clades of angiosperms, including *Amborella trichopoda* (Fig. 7b and Supplementary Figs. 6 and 7), which has branched from the basal angiosperm lineage[42,43]. This finding suggests that PnsB2 originated via gene duplication of *NDF5* in the early evolution of angiosperms. This is an example of the evolution of a novel subunit of the protein complex from an assembly factor required for complex formation.

**PnsB3 is likely to have evolved from a protein unrelated to NDH.** One of the linker proteins, Lhca6, was also acquired in angiosperms after branching from the bryophyte lineage[28], as was PnsB2. SubB is the contact site for Lhca6[32]. A dramatic evolutionary remodeling likely occurred in SubB to establish the interaction with Lhca6 in a common ancestor of angiosperms. To study this SubB remodeling more extensively, we compared the other SubB subunits in Arabidopsis to those in Marchantia and Physcomitrella using reciprocal Blastp analyses. Protein sequences showing similarity with Arabidopsis PnsB1, PnsB4, and PnsB5 were found in both Marchantia and Physcomitrella (Fig. 4a). Each top hit sequence returned to the original Arabidopsis protein, respectively, in reverse-Blastp (Fig. 4a and Supplementary Tables 1 and 2), suggesting that PnsB1, PnsB4, and PnsB5 exist in both bryophytes. Consistent with this result, Physcomitrella mutants with the *Pp3c16_3930V3* gene knocked

out lacked NDH activity, suggesting that the gene encodes the NDH subunit PnsB4 in Physcomitrella[28]. The PnsB1 protein was also detected using the antibody against Arabidopsis PnsB1 in Marchantia and Physcomitrella[20,28] (Fig. 5c). The PnsB1 protein is likely encoded by two genes in the Physcomitrella genome (Supplementary Table 2), probably because of whole-genome duplication[44]. However, no hits against PnsL3 with an E-value of less than $1.00E^{-10}$ were found in the Marchantia protein database (Fig. 4a and Supplementary Table 1). Pp3c5_4110V3 of Physcomitrella is faintly similar to Arabidopsis PnsL3, but the protein is likely an ortholog of PsbQ-1 or PsbQ-2 of Arabidopsis (Fig. 4a and Supplementary Table 2). PsbQ proteins are subunits of PSII, while PnsL3 is paralogous to them in Arabidopsis[45]. It seems that the *PnsL3* gene does not exist in these bryophytes and was acquired from the *PsbQ* gene in a common ancestor of angiosperms. Some SubB subunits and Lhca6 can be assembled in the absence of PnsL3[32], suggesting that PnsL3 is not directly required to establish the interaction between SubB and Lhca6. On the basis of this fact, we do not focus on PnsL3 as a candidate for the critical evolutionary step for establishing the interaction with Lhca6.

Mapoly0075s0083 in Marchantia and Pp3c14_14470V3 in Physcomitrella were hit by Arabidopsis PnsB3 as a query (Fig. 4a). Hereinafter, these proteins and genes in bryophytes are called Protein Y and *Gene Y*, respectively. However, Protein Y is more similar to an unknown protein encoded by the *AT4G32590* gene in Arabidopsis than PnsB3. AT4G32590 is a protein that resembles PnsB3 (hereinafter we call it 'PnsB3-like'), but it has been reported that *AT4G32590* is not required for NDH activity[34]. We also confirmed that the Arabidopsis mutant defective in *AT4G32590* showed normal NDH activity and the PnsB1 accumulation (Supplementary Fig. 8), suggesting that PnsB3-like is not related to the NDH complex. PnsB3-like was most similar to Protein Y in Marchantia and Physcomitrella according to Blastp analysis (Fig. 4a and Supplementary Table 3). In the phylogenetic analysis, Protein Y was included in the PnsB3-like clade (Supplementary Fig. 9b). Protein Y may not be related to the NDH complex in bryophytes. To assess this possibility, we knocked out *Gene Y* in Physcomitrella by replacing the gene with the *Aph4* cassette (Supplementary Fig. 4c). Genomic PCR confirmed the *Gene Y* knockout (Supplementary Fig. 4d). The *Gene Y* expression was also below the detection limit in RT-PCR (Fig. 8a). The knockout lines did not show any

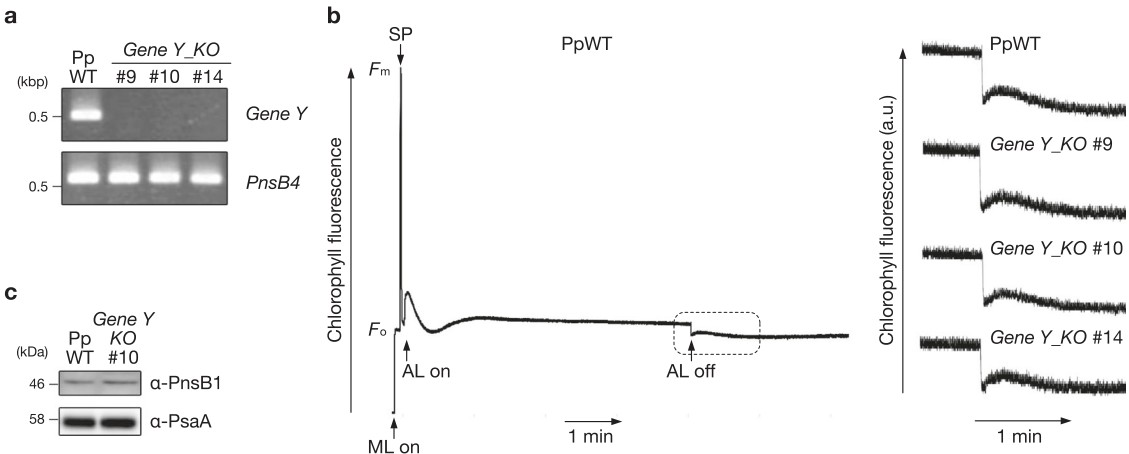

**Fig. 8 The NDH activity was not affected in the *Gene Y* knockout mutant in Physcomitrella. a** RT-PCR analysis of *Gene Y* and *PnsB4* transcripts. These transcripts were amplified using cDNA from Physcomitrella wild-type (PpWT) and *Gene Y* knockout mutants (*Gene Y_KO*). *PnsB4* was detected as a control. **b** Analysis of transient increases in chlorophyll fluorescent after turning off actinic light (AL), as in Fig. 5b. A typical trace of chlorophyll fluorescent in the PpWT and the transient increases in chlorophyll fluorescent (boxed area) are shown at left and right, respectively. **c** Immunoblot analysis of membrane proteins from PpWT and *Gene Y_KO* #10. Samples were loaded on the basis of chlorophyll content, and PnsB1 and PsaA (a loading control) were detected. Each experiment was performed at least twice, and results in independent transgenic plants are also shown in Supplementary Fig. 11d.

mutant phenotypes in their growth (Supplementary Fig. 4f), and NDH activity and PnsB1 accumulation were at the same levels as seen in the WT (Fig. 8b, c). Protein Y is not required for the NDH complex in Physcomitrella. Most likely, Protein Y is orthologous to PnsB3-like, although their function is not clear in any plant species. No other proteins were hit against PnsB3 in either bryophyte. Bryophytes do not contain PnsB3, as in the case of PnsB2. PnsB3 may have originated from Protein Y/AT4G32590 during the evolution of angiosperms and the SubB remodeling for the Lhca6-dependent supercomplex formation with PSI-LHCI.

## Discussion

Like mitochondrial Complex I, consisting of 44 subunits in mammals[46], the chloroplast NDH complex consists of about 30 subunits in angiosperms[11,47], whereas the respiratory and photosynthetic NDH complex in proteobacteria and cyanobacteria consists of 14 and 18 subunits, respectively[7,8,48]. Both types of NDH complex have increased in size during the evolution of eukaryotes. It seems that mitochondrial Complex I obtained its huge structure immediately after the birth of eukaryotic organisms, i.e., in the Last Eukaryotic Common Ancestor (LECA), because many eukaryote-specific subunits are widely conserved in mammals, plants, and Trypanosoma[49]. In contrast, the molecular size of the chloroplast NDH complex has gradually increased during the evolution of land plants. Our results suggest that PnsB2 and PnsB3 are not conserved in bryophytes, and are most likely to have been acquired from a common ancestor of angiosperms. It is also thought, based on Blast search[20,50], that the SubL subunits (PnsL1–PnsL4) are not conserved in bryophytes, but are specific to angiosperms. Moreover, the chloroplast NDH complex in angiosperms acquired the novel supercomplex formation via a copy of PSI-LHCI depending on Lhca6[20,28].

Our results suggest that Protein X in Physcomitrella can work as NDF5 in Arabidopsis and is an ortholog of NDF5 (Fig. 6). The results also suggest that a novel chloroplast NDH subunit, PnsB2, originated from Protein X/NDF5 via tandem gene duplication (Fig. 7 and Supplementary Figs. 6 and 7). An assembly factor-like subunit (NDUFA12) or a subunit-like assembly factor (NDUFAF2) has also been reported in respiratory Complex I[51,52]. NDUFA12 and NDUFAF2 show sequence similarity, as observed between PnsB2 and NDF5, but it was previously unclear which

protein is ancestral. In this study, we have clarified the order of evolution: in this case, a subunit was evolved from an assembly factor. In addition to PnsB2, there are several photosynthetic NDH-specific subunits derived from similar origins. For example, PnsL4 and PnsL5 (also known as FKBP16-2 and CYP20-2, respectively) subunits are members of the immunophilin family[25,53]. Generally, immunophilins assist protein folding and/or act as chaperones[54]. NdhT and NdhU (also known as CRRJ and CRRL, respectively) subunits have J and J-like domains, respectively[9]. J proteins were originally identified as co-chaperones of 70 kDa heat shock proteins[55]. It may not be unusual for a novel subunit of a protein complex to be recruited from a protein that supports protein folding or assembly.

NDF5 was required to initiate the early stage of SubB assembly in Arabidopsis (Supplementary Fig. 3). Since PnsB2 and NDF5 are paralogous, one expectation is that PnsB2 is put in place of NDF5 once the assembly process of SubB has proceeded to a certain stage. However, this scenario is unlikely because PnsB2 is a component of the NDF5 complex, as are PnsB3 and Lhca6 (Fig. 2). Subsequently, this initial assembly intermediate incorporates PnsB4, PnsB5 and also, most likely, CRR3. Incorporation of PnsL3 and PnsB1 occurs in a later stage of SubB assembly[32]. Because bryophytes lack PnsB2, PnsB3, or Lhca6, we speculate that Protein X/NDF5 in bryophytes and angiosperms may assist the assembly of other SubB subunits such as PnsB4 and PnsB5.

PnsB2, PnsB3, and Lhca6 are necessary to form the initial assembly intermediate of SubB. Other SubB subunits are not needed (Fig. 2), suggesting that PnsB2 and PnsB3 form the binding site of Lhca6. As described above, the Lhca6-dependent NDH-PSI supercomplex formation was acquired in a common ancestor of angiosperms[28]. Lhca6 originated from Lhca2, which is one of the antenna proteins of PSI (LHCI). Modification of the stromal loop of Lhca2 was necessary for the evolutionary acquisition of its linker function[30]. In this study, we clarified the evolutionary scenario for the supercomplex formation on the NDH side, in which novel subunits were established in SubB. An assembly factor, Protein X/NDF5, was modified to form a new subunit, PnsB2, after tandem gene duplication. In an amino acid alignment, a PnsB2-specific insertion was found at its C-terminal region (Supplementary Fig. 10), supporting that neofunctionalization occurred in PnsB2. A fern, *Selaginella moellendorffii*, encodes a protein locating at the same clade with bryophyte

Protein Xs in the phylogenetic tree (Supplementary Fig. 9a), and no other proteins showing homology with PnsB2 or NDF5 were found. Whereas some gymnosperms lost the NDH complex[56], the gene duplication and neofunctionalization may have occurred in the common ancestor of angiosperms, since the tandem arrangement of the NDF5 and PnsB2 genes are still conserved in various clades of angiosperms including Amborella (Supplementary Fig. 7).

PnsB3 appears to have originated from Protein Y/AT4G32590, although we do not know the function of the original protein. Notably, the SubB assembly in angiosperms is initiated by PnsB2, PnsB3, and Lhca6, which were newly acquired during the evolution of angiosperms. Our discoveries suggest that the assembly process of the NDH complex was modified to permit Lhca6-dependent supercomplex formation. Lhca6 irregularly forms a heterodimer with Lhca3 instead of Lhca2 for the Lhca6-dependent NDH-PSI interaction[31] (Supplementary Fig. 1b). We speculate that the attachment of PnsB2 and PnsB3 stabilizes the Lhca3/Lhca6 heterodimer specifically localized to the NDH-PSI supercomplex. Because the monomeric NDH complex is fully assembled in the lhca5 lhca6 double mutant[25], the modification of the assembly process was for the NDH-PSI supercomplex rather than the NDH complex.

Why did angiosperms acquire the Lhca6-dependent NDH-PSI supercomplex formation? Previously, we hypothesize that it was an adaptation to terrestrial light environments on the basis of the mutant phenotype[5,28]. Supercomplex formation was necessary to stabilize the NDH complex, especially under excessive light conditions ($>500 \mu mol$ photons $m^{-2} s^{-1}$)[5,28]. Even under growth chamber conditions ($50-100 \mu mol$ photons $m^{-2} s^{-1}$), the accumulation level of NDH was lower in mature leaves than newly developing leaves in the Arabidopsis lhca6 mutant[25]. In angiosperms, the Lhca6-dependent NDH-PSI supercomplex formation is necessary for the long-term stability of the NDH complex even in medium-intensity light environments. On the basis of the leaf stage-specific expression of assembly factors (Fig. 3), the NDH complex is likely preferentially synthesized in immature leaves and is not actively synthesized in mature leaves in Arabidopsis. Once the leaves are fully expanded in angiosperms, they are used for long periods of a month to a few years[57]. Instead of continuously supplementing the NDH complex in mature leaves, angiosperms selected to stabilize the NDH complex by formation of the Lhca6-dependent supercomplex.

## Methods

**Plant materials and growth conditions**. Arabidopsis (Arabidopsis thaliana; Columbia-0) was grown in soil in a growth chamber ($50 \mu mol$ photons $m^{-2} s^{-1}$, 16 h photoperiod, 22 °C) for 4 weeks. The T-DNA line Salk_009697C was provided by the Salk Institute Genomic Analysis Laboratory. Physcomitrella patens Bruch & Schimp subsp. patens was used in this study. Protonemata of Physcomitrella were cultured on BCDAT agar medium[58] with 0.5% glucose at 25 °C under constant light ($50 \mu mol$ photons $m^{-2} s^{-1}$). For immunoblot analysis, homogenized protonemata were shaken in BCDAT liquid medium with 0.5% glucose for 10–12 days under the same conditions.

**Arabidopsis transformation**. The coding sequence of Gene_X was amplified from Physcomitrella cDNA by PCR (the primers used in this study are listed in Supplementary Table 4). 5′ CUTR of NDF5 and a terminator of HSP18.2 were amplified from Arabidopsis genomic DNA and fused to the 5′ and 3′ sides of the Gene_X PCR product by second PCR, respectively. The resultant PCR product was cloned into the NotI and AscI digested pENTR/D-TOPO (Invitrogen) via an In-Fusion system (TaKaRa Bio). The chimeric gene was confirmed by sequencing and then introduced into the binary vector pGWB2[59] by LR Clonase reaction (Invitrogen). The resulting plasmid was introduced into Agrobacterium tumefaciens C58 by electroporation, and the bacteria were used to transform the Arabidopsis ndf5 and pnsb2 mutants.

**Knockout of Gene X and Gene Y in Physcomitrella**. For knockout of Gene X, a 1.2 kb upstream region and a 1.0 kb downstream region were amplified and inserted via the In-Fusion system into the EcoRV and XmaI sites of pTN186 (GenBank: AB542059), respectively; the two cloning sites were arranged on either side of the aph4 cassette. For knockout of Gene Y, a 1.0 kb upstream region and a 1.0 kb downstream region were amplified and inserted into the EcoRV and XmaI sites of pTN186, respectively. The resultant plasmids were linearized before introduction into Physcomitrella protoplast cells. For transformation, protoplasts were prepared by treating protonemal cells with 2% Driselase (Kyowa Hakko, Japan) in 8% (w/v) mannitol, and were suspended in a buffer (8.3% (w/v) mannitol, 15 mM MgCl$_2$, 0.1% MES (pH 5.6)). The linearized plasmids and equal volume of PEG solution (28.5% (w/v) PEG6000, 100 mM Ca(NO$_3$)$_2$, 10 mM Tris-HCl (pH 8.0), 7.2% mannitol) were mixed with the protoplasts. After incubation at 45 °C for 5 min and subsequent 20 °C for 10 min, cell suspension was diluted stepwise with a recovery medium (5 mM Ca(NO$_3$)$_2$, 1 mM MgSO$_4$, 0.045 mM FeSO$_4$, 0.18 mM KH$_2$PO$_4$ (pH 6.5), Alternative TES[58], 5 mM ammonium tartrate, 6.6% (w/v) mannitol, 0.5% (w/v) glucose), and then incubated in the dark at 25 °C for 1 day. After the recovery, the cells were resuspended in PRM/T (BCD medium supplemented with 5 mM ammonium tartrate, 10 mM CaCl$_2$, 8% (w/v) mannitol, 0.8% (w/v) agar) prewarmed at 45 °C, and then spread on PRM/B (BCD medium supplemented with 5 mM ammonium tartrate, 10 mM CaCl$_2$, 6% (w/v) mannitol, 0.8% (w/v) agar). After 5 days, cells were transferred to BCDAT medium containing hygromycin B for the selection of transformants.

**Extraction of nucleic acid, cDNA synthesis, and quantitative PCR**. Genomic DNA in Physcomitrella was extracted using the cetyl trimethyl ammonium bromide method[60]. Total RNA was isolated from leaves in Arabidopsis or protonemata in Physcomitrella using a Maxwell 16 Instrument and Maxwell LEV Plant RNA Kit (Promega). DNase I treatment was included in the Maxwell system. Single-stranded cDNA was synthesized from 750 ng of RNA using ReverTra Ace qPCR RT Master Mix (Toyobo).

Quantitative real-time PCR was performed as described[61]. FastStart SYBR Green Master (ROX; Roche) and a MX3000P system were used in accordance with the manufacturer's instructions. Quantitative estimations were made with MxPro™ software using the ΔΔCt (cycle threshold) method (Stratagene, Agilent Technologies). The EF1α gene in Arabidopsis was used as an internal control, and the data were calibrated using the 35Sp::Gene X/ndf5 lines #23 and #12 as 1 in Fig. 6b and Supplementary Fig. 6b, respectively.

**Monitoring of transient increases in chlorophyll fluorescence**. Transient increases in chlorophyll fluorescence after turning off the actinic light ($50 \mu mol$ photons $m^{-2} s^{-1}$, white light for 5 min) were monitored using a MINI-PAM portable chlorophyll fluorometer (Waltz)[39]. The Fm level was recorded by applying a saturating pulse of white light (800 ms at 3000 $\mu mol$ photons $m^{-2} s^{-1}$) and was used for standardizing the fluorescence levels. Physcomitrella gametophores grown on BCDAT agar medium with 0.5% glucose for 13 days and Arabidopsis plants grown on soil before bolting were monitored in ambient air.

**Antibody preparation**. cDNA encoding the Arabidopsis NDF5 without its chloroplast transit peptide predicted by ChloroP 1.1[62] was amplified. The amplified sequence was digested with AseI and XhoI and cloned into NdeI and XhoI digested pET-22b(+) (Novagen) using Ligation high (Toyobo). Expression of the recombinant proteins was induced by 1 mM isopropyl β-D-thiogalactopyranoside at 37 °C for 4 h in host Escherichia coli strain Rosetta (DE3) pLysS cells (Novagen). After induction, the cells were harvested in 20 mM potassium phosphate buffer (pH 7.4) containing 40 mM imidazole, 500 mM NaCl, and cOmplete™ EDTA-free protease inhibitor cocktail (Roche). The inclusion bodies were pelleted from sonicated cells at 3000g for 15 min and solubilized in 20 mM potassium phosphate buffer (pH 7.4) containing 40 mM imidazole, 500 mM NaCl, and 6 M guanidine hydrochloride. Insoluble material was removed by centrifugation at 48,000g for 1 h. The supernatant was incubated with Ni-NTA Agarose (Qiagen) for 1 h. The Ni-NTA Agarose was washed with 20 mM potassium phosphate buffer (pH 7.4) containing 40 mM imidazole, 500 mM NaCl, and 4 M urea. The recombinant proteins were eluted with 20 mM potassium phosphate buffer (pH 7.4) containing 500 mM imidazole, 500 mM NaCl, and 4 M urea. Polyclonal antisera were raised against the purified recombinant protein in a mouse (T. K. Craft, Maebashi, Japan).

**Chloroplast membrane preparation in Arabidopsis, total membrane preparation in Physcomitrella, and immunoblot analysis**. Chloroplasts in Arabidopsis were isolated as described[17] with minor modifications. Leaves were homogenized in 20 mM Tricine-NaOH (pH 7.6) containing 330 mM sorbitol, 5 mM EGTA, 10 mM Na$_2$CO$_3$, 10 μM E-64, and 100 μM leupeptin. Chloroplasts were pelleted at 2000g for 5 min. To prepare chloroplast membranes, chloroplasts were ruptured in 20 mM HEPES-KOH (pH 7.6) containing 5 mM MgCl$_2$, 2.5 mM EDTA, 10 μM E-64, and 100 μM leupeptin. Chloroplast membranes were separated by centrifugation at 20,000g for 2 min. Total membranes in Physcomitrella were prepared as previously described[28]. The chlorophyll content was calculated according to the method of Porra et al.[63]. For immunoblot analysis, 14% or 16% acrylamide gels were used. Loading volume was based on the chlorophyll content. Signals were detected with ECL Prime Western Blotting Detection Reagent (GE Healthcare) and visualized with LAS4000 (GE Healthcare). Polyclonal antibody

against PsaA was purchased from Agrisera (AS06 172) and used at a dilution of 1:10,000. Polyclonal antibody against PnsB1[34], PnsB2[34], NdhH[64], and Cytf[65] were used at a dilution of 1:5000. Polyclonal antibody against NDF5, PnsB3[66], PnsB4[36], PnsB5[25], and PnsL3[45] were used at a dilution of 1:2000. Polyclonal antibody against CRR3[32] was used at a dilution of 1:500. Secondary antibody against rabbit IgG (Anti-rabbit IgG, HRP-Linked Whole Ab Donkey (NA934, Cytiva)) was used at a dilution of 1:15,000 for the detection of PnsB3, PnsB4, and CRR3 and at a dilution of 1:25,000 for the detection of PsaA, PnsB1, PnsB2, PnsB5, PnsL3, and NdhH. Secondary antibody against mouse IgG (Anti-mouse IgG, HRP-Linked Whole Ab sheep (NA931, Cytiva)) was used at a dilution of 1:25,000 for the detection of NDF5. For Coomassie Brilliant Blue (CBB) staining, Bio-Safe Coomassie stain (Bio-Rad) or CBB Stain One Super (Nacalai) were used after SDS-PAGE.

**Protein complex separation**. BN-PAGE was performed as previously described[67] with minor modifications. Arabidopsis chloroplast membranes were washed twice with 25 mM BisTris-HCl (pH 7.0) containing 20% (w/v) glycerol and then solubilized in 25 mM BisTris-HCl (pH 7.0) containing 20% (w/v) glycerol and 1% (w/v) n-dodecyl-β-D-maltoside for 5 min on ice. The concentration of chlorophyll was adjusted to 1.0 μg μL$^{-1}$. Insoluble materials were removed by centrifugation at 20,000g for 2 min. Supernatants were mixed with one-tenth volume of 100 mM BisTris-HCl (pH 7.0) containing 500 mM 6-aminocaproic acid, 30% sucrose, and 5% SERVA Blue G, and protein complexes equivalent to 10 μg of chlorophyll were separated by 5–12% acrylamide gradient gel containing 50 mM BisTris-HCl (pH 7.0) and 500 mM 6-aminocaproic acid. After electrophoresis, BN-gel was stained with Bio-Safe Coomassie stain (Bio-Rad).

SDG was performed as previously described[32]. Young leaves (Fig. 3a, stages 1 and 2) were used. Chloroplast membranes were washed once with 5 mM Tricine-NaOH (pH 8.0) containing 10 μM E-64 and 100 μM leupeptin and then solubilized with 5 mM Tricine-NaOH (pH 8.0) containing 0.9% (w/v) n-dodecyl-β-D-maltoside, 10 μM E-64, and 100 μM leupeptin. The chlorophyll concentration was adjusted to 1.0 mg mL$^{-1}$. Chloroplast membranes were dissolved for 5 min on ice and then loaded on the top of the sucrose gradient (5–40%) prepared with 25 mM MES-NaOH (pH 6.8) containing 5 mM MgCl$_2$, 10 mM NaCl, 0.02% (w/v) n-dodecyl-β-D-maltoside, 10 μM E-64, and 100 μM leupeptin. The protein complexes were separated by ultracentrifugation for 24 h by using an SW32.1-Ti rotor (Beckman) at 28,700 rpm (150,000g). After ultracentrifugation, the gradients were fractionated into 30 fractions using Gradient Station (BIOCOMP). Proteins from equal amounts of fractions were precipitated by adding a one-fifth volume of 100% (w/v) trichloroacetic acid and centrifuging at 20,000g for 5 min. Pellets were washed twice with 99% (v/v) ice-cold acetone, dissolved in 1x Laemmli buffer, and subjected to further SDS-PAGE and immunoblot analysis. The photos of the SDG tube and the SDS-PAGE lanes were aligned according to the major green bands (LHCII, PSII monomer, and PSI-LHCI) and the pattern of the CBB-stained gels (Supplementary Fig. 2b).

**Phylogenetic analysis**. Sequences of orthologous proteins against Arabidopsis NDF5 and PnsB2 in angiosperms were retrieved from Phytozome V12 (https://phytozome.jgi.doe.gov/pz/portal.html#) using Blastp search. It was confirmed that each top hit sequence returned to the Arabidopsis NDF5 and PnsB2, respectively, as the best hit in reverse-Blastp. The obtained NDF5 or PnsB2 sequences were aligned independently using ProbCons[68]. The chloroplast transit peptides were excluded using Extractalign included in the EMBOSS package[69] based on those positions of Arabidopsis NDF5 or PnsB2 predicted by ChloroP 1.1. The chloroplast transit peptides of Protein X in Marchantia, Physcomitrella, and Selaginella were predicted by ChloroP 1.1 and excluded. The sequences of NDF5, PnsB2, and Protein X were combined in a FASTA file and aligned using ProbCons. A Bayesian inference was performed using MrBayes version 3.2.6[70] using the WAG model and gamma-distributed rate variation. One and three million generations were completed for Fig. 4b and Supplementary Fig. 9a, respectively, and trees were collected every 100 generations, after discarding trees corresponding to the first 25% (burn-in), to generate a consensus phylogenetic tree. Bayesian posterior probabilities were estimated as the proportion of trees sampled after burn-in.

Construction of a phylogenetic tree of PnsB3, PnsB3-like, and Protein Y was performed by close to the same procedure described above. In MrBayes, five million generations were completed, and trees were collected every 100 generations, after discarding trees corresponding to the first 25% (burn-in), to generate a consensus phylogenetic tree. Bayesian posterior probabilities were estimated as the proportion of trees sampled after burn-in.

The protein sequences used in the phylogenetic analysis can be found in the Phytozome V12 database under the accession numbers listed in Fig. 4, Supplementary Fig. 7 and Supplementary Table 5.

**Reporting summary**. Further information on research design is available in the Nature Research Reporting Summary linked to this article.

## Data availability
The data that support the findings of this study are available in Source Data file or from the corresponding author upon reasonable request. Source data are provided with this paper.

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

## Acknowledgements

The authors thank Tsuyoshi Endo for lines *ndf5*, *pnsb1*, *pnsb2*, *pnsb3*, and *pnsb4* and for antisera against PnsB1, PnsB2, PnsB3, and PnsB4; Kentaro Ifuku for line *pnsl3* and for antisera against PnsL3; Hualing Mi for antisera against NdhH; Amane Makino for antisera against Cyt*f*; and Mitsuyasu Hasebe for the pTN186 plasmid. This work was supported by the Japan Society for the Promotion of Science (16H06555 and 19H00992 for T.S.) and as a research fellow (17J09745 and 19J01563 for Y.K.).

## Author contributions

Y.K. and T.S. designed the research. Y.K. performed almost all the experiments. M.O. constructed the Physcomitrella mutants. Y.K. and T.S. prepared the article.

## Competing interests

The authors declare no competing interests.

**Additional information**

