## [Peer Review File. · Nature Communications]

REVIEWER COMMENTS

Reviewer #1 (Remarks to the Author):

Alternative electron transport pathways contribute in the regulation of photosynthetic electron transport. This manuscript describes the search (in *Physcomitrella patens*) and the characterization (in *Physcomitrella patens* and *Arabidopsis thaliana*) of proteins involved in the formation and stabilization of chloroplast NADH dehydrogenase-like complex with Photosystem I (to form NDH-PSI supercomplex). A special attention was given to the linker protein Lhca6 and to NDH subcomplex B. The study of NDH complex has several evolutionary implications since it has prokaryotic origins, it is structurally related to mitochondrial Complex I and its complexity is increasing along plant evolution and adaptation to land environment.

In particular, the authors characterized in *Arabidopsis* the role of the assembly factor NDF5 (NDH-dependent Cyclic Electron Flow 5). NDF5 initiates the assembly of NDH subunits (PnsB2 and PnsB3) and Lhca6. They identified an orthologous gene for PnsB2 or NDF5 in Bryophytes, its functional characterization in *Physcomitrella patens* allowed to demonstrate that it has a role in NDH activity and it can complement the *Arabidopsis* *ndf5* mutant. Viceversa the search for an orthologue for PnsB3 in bryophyte did not end into the identification of its ancestor.

This is an excellent manuscript, describing novel information. Quality of experimental results is high and the text is written clearly. Here you are some points to be considered before publication.

- Concerning the evolutionary history of NDF5/PsnB2 and PsnB3 the authors first focus on the search of gene x and then on gene y. I wonder whether there could be an advantage to present the search of gene x and y together and then to move to the characterization of gene Y to close with the nice positive results obtained on gene x (this is just a suggestion).
- I think that in the main body of the manuscript there is the need for a stronger phylogenetic analysis, rather than "simple" reverse-blast p searches. The goal is to trace the evolution of NDH-PSI supercomplex formation and therefore the phylogenetic analysis is central.
- Page 9, Figure 1 and Supplementary Figure 3, I would expect to see the Immunoblot results for PsnB3, PsnB4 and PsnL3 not only for mutant lines but also for WT. They could also possibly be added into the Supplementary files. Aren't they important control for the mutant lines?
- Information contained in Figure 2 should arrive before those delivered by figure 1. It is useful to know first the total amount of protein accumulation and then their distribution along the sucrose gradient.
- Figure 3c. Immunoblotting of other assembly factors (e.g. *crr3*) or other NDH-1 complex proteins (e.g. NDHH, NDHM, etc...) would be useful to support alpha-NDF5 and alpha-PNSB2 immunoblot profile.
- Supplementary Figure 4a is an interesting piece of information, why not adding it in Figure 4?
- Is protein x present in *Physcomitrella* proteome? A specific antibody against gene X was not obtained but can protein x/NDF5 be detected by mass spectrometry? Can the authors look in their previous proteomic experiments? Anyway, functional complementation experiments is a good tool to verify the role of *Physcomitrella patens patens* gene X.
- A deeper physiological characterization of the mutants (not easy in the case of NDH complex maybe?) would make stronger the considerations about adaptation to terrestrial light environments based on the mutant phenotype (page 27).
- The methods followed to monitoring the transient increases in chlorophyll fluorescence should be better described (e.g. what is the amount of light? what light quality, anoxia?), what medium was used for plant growth? Is this method enough for physiological considerations (see previous point)?

Reviewer #2 (Remarks to the Author):

The manuscript by Kato et al. deals with the formation of the NDH-PSI supercomplex in *Arabidopsis* and *Physcomitrella*. In the first part of the paper, the authors show that assembly of the supercomplex depends on the protein NDF5, which is not part of the final complex. Instead, it can be detected specifically in a smaller assembly that seems to accumulate in mutants of the NDH

subB complex. This complex contains the subunits PnsB2/B3 and the corresponding mutants do not assemble the NDH5 containing complex. The second part of the paper deals with a NDH5/PnsB2 ortholog in *Physcomitrella*. The corresponding mutant is deficient in NDH activity and expression of the gene in *Arabidopsis* mutants indicates complementation of NDH5 but not PnsB2. The authors concluded from this data that the PnsB2 subunit evolved from the potential assembly factor NDH5. The work is interesting, but multiple issues raised:

Firstly, by comparing the presented work with the paper of Ishida et al. (<https://doi.org/10.1093/pcp/pcn205>), the novelty is rather low. Ishida et al. discovered NDH5 in *Arabidopsis* and showed that it is essential for accumulation of active NDH complexes. Moreover, they have identified already the NDH5 ortholog in *Physcomitrella*.

Secondly, the NDH5 containing complex was not really defined biochemically. The authors observed comigration of selected subB components with NDH5 by sucrose density centrifugation and immunoblot analysis, but the biochemical nature of this complex remained unclear. The manuscript needs more data that show the composition of this complex (e.g. by mass spectrometry based complexome profiling, specific isolation of the complex or other techniques).

Thirdly, almost all data are based on single experiments or at least only data of single experiments were shown. The authors should add a thorough statistical evaluation of the results, e.g. for analysis of the NDH5 complementation mutant. How significant are the differences in the fluorescence signals? What is the variation between single measurements? The same applies in principle for the immunoblot data.

Fourthly, the characterization of *Physcomitrella* protein X is very preliminary. The *Physcomitrella* complementation mutant is missing and no data was presented that corroborated the function in this organism.

Responses to comments by reviewers

Reviewer #1 (Remarks to the Author):

The authors answered to all questions and requests.

Reviewer #2 (Remarks to the Author):

The authors addressed my major concern about the reproducibility of results and added independent data for Figures 1, 3, 5, and 8 (Supplementary Figure 11). I still believe that a more thorough statistical evaluation and a deeper biochemical characterization of the complex could have improved the manuscript substantially. However, I recognize the technical difficulties and agree in principle with the authors' statements.

Thank you for your positive evaluation and understanding. Your suggestions are very helpful for improving the manuscript.

REVIEWERS' COMMENTS

Reviewer #1 (Remarks to the Author):

The authors answered to all questions and requests.

Reviewer #2 (Remarks to the Author):

The authors addressed my major concern about the reproducibility of results and added independent data for Figures 1, 3, 5, and 8 (Supplementary Figure 11). I still believe that a more thorough statistical evaluation and a deeper biochemical characterization of the complex could have improved the manuscript substantially. However, I recognize the technical difficulties and agree in principle with the authors' statements.

Responses to comments by reviewers

Reviewer #1 (Remarks to the Author):

Alternative electron transport pathways contribute in the regulation of photosynthetic electron transport. This manuscript describes the search (in *Physcomitrella patens*) and the characterization (in *Physcomitrella patens* and *Arabidopsis thaliana*) of proteins involved in the formation and stabilization of chloroplast NADH dehydrogenase-like complex with Photosystem I (to form NDH-PSI supercomplex). A special attention was given to the linker protein Lhca6 and to NDH subcomplex B. The study of NDH complex has several evolutionary implications since it has prokaryotic origins, it is structurally related to mitochondrial Complex I and its complexity is increasing along plant evolution and adaptation to land environment.

In particular, the authors characterized in *Arabidopsis* the role of the assembly factor NDF5 (NDH-dependent Cyclic Electron Flow 5). NDF5 initiates the assembly of NDH subunits (PnsB2 and PnsB3) and Lhca6. They identified an orthologous gene for PnsB2 or NDF5 in Bryophytes, its functional characterization in *Physcomitrella patens* allowed to demonstrate that it has a role in NDH activity and it can complement the *Arabidopsis ndf5* mutant. Viceversa the search for an orthologue for PnsB3 in bryophyte did not end into the identification of its ancestor.

This is an excellent manuscript, describing novel information. Quality of experimental results is high and the text is written clearly. Here you are some points to be considered before publication.

- Concerning the evolutionary history of NDF5/PsnB2 and PsnB3 the authors first focus on the search of gene x and then on gene y. I wonder whether there could be an advantage to present the search of gene x and y together and then to move to the characterization of gene Y to close with the nice positive results obtained on gene x (this is just a suggestion).

--- Thank you for your advice. Indeed, we wrote the text as you suggest in the original draft, but we found it rather discursive. Finally, we selected the present order.

- I think that in the main body of the manuscript there is the need for a stronger phylogenetic analysis, rather than “simple” reverse-blast p searches. The goal is to trace the evolution of NDH-PSI supercomplex formation and therefore the phylogenetic analysis is central.

--- We are grateful for your advice. We added the fern, *Selaginella moellendorffii*, in the phylogenetic tree of Protein X, NDF5, and PnsB2 and discussed when *Gene X* duplication and PnsB2 neofunctionalization occurred. We also found a PnsB2-specific insertion in its amino acid sequence in angiosperms, supporting that PnsB2 was neofunctionalized after duplication of the ancestral gene.

- Page 9, Figure 1 and Supplementary Figure 3, I would expect to see the Immunoblot results for PsnB3, PsnB4 and PsnL3 not only for mutant lines but also for WT. They could also possibly be added into the Supplementary files. Aren't they important control for the mutant lines?

--- In SDG fractions of the WT, the major signals are derived from the NDH-PSI supercomplex, which is much more abundant than NDH assembly intermediates. The signal of the assembly intermediate is probably masked by the tail of the major signal, as is the pattern by the PnsB2 antibody. Actually, we observed the similar pattern of PnsB1 in the SDG of WT plants in a previous study (Figure 2A in Kato et al., 2018, *Plant Physiology*).

- Information contained in Figure 2 should arrive before those delivered by figure 1. It is useful to know first the total amount of protein accumulation and then their distribution along the sucrose gradient.

--- We are grateful for your advice. Now, we showed the accumulation levels of NDF5 and other proteins in Figure 1.

- Figure 3c. Immunoblotting of other assembly factors (e.g. *crr3*) or other NDH-1 complex proteins (e.g. NDHH, NDHM, etc...) would be useful to support alpha-NDF5 and alpha-PNSB2 immunoblot profile.

--- We added the results of CRR3 and NdhH.

- Supplementary Figure 4a is an interesting piece of information, why not adding it in Figure 4?

--- Thank you for your advice. We added the simple phylogenetic tree of Protein X, NDF5, and PnsB2 using representative angiosperm species to make it easy to see, and the detail phylogenetic tree including Selaginella was also shown in Supplementary Fig. 9a.

- Is protein x present in Physcomitrella proteome? A specific antibody against gene X was not obtained but can protein x/NDF5 be detected by mass spectrometry? Can the authors look in their previous proteomic experiments? Anyway, functional complementation experiments is a good tool to verify the role of Physcomitrella patens gene X.

--- Due to the fragile nature of the NDH-PSI supercomplex after solubilization in Physcomitrella, only limited number of NDH and PSI subunits were detected in the previous Co-IP/MS experiments (Kato et al., 2018, Plant J.). Protein X was not detected in this proteomics study, although we are unsure the reason.

- A deeper physiological characterization of the mutants (not easy in the case of NDH complex maybe?) would make stronger the considerations about adaptation to terrestrial light environments based on the mutant phenotype (page 27).

--- We are sorry for confusing. We discussed about adaptation to terrestrial light environments based on the mutant phenotype in the previous study (not in this study). We tried to discuss that the Lhca6-dependent supercomplex formation is necessary for long-term stability of the NDH complex, and plants do not need highly active de novo synthesis of the NDH complex in mature leaves supported by leaf age-dependent accumulation of SubB assembly factors (Figure 3). We slightly rephrased that paragraph to make it easier to read.

- The methods followed to monitoring the transient increases in chlorophyll fluorescence should be better described (e.g. what is the amount of light? what light quality, anoxia?), what medium was used for plant growth? Is this method enough for physiological considerations (see previous point)?

--- We added the explanation about the light quality and the air condition in addition to light intensity in Methods. We also described the plant growth conditions. This measurement provides whether plants possess NDH activity but is not quantitative, so we hesitate to discuss physiological aspect from this measurement.

Reviewer #2 (Remarks to the Author):

The manuscript by Kato et al. deals with the formation of the NDH-PSI supercomplex in *Arabidopsis* and *Physcomitrella*. In the first part of the paper, the authors show that assembly of the supercomplex depends on the protein NDF5, which is not part of the final complex. Instead, it can be detected specifically in a smaller assembly that seems to accumulate in mutants of the NDH subB complex. This complex contains the subunits PnsB2/B3 and the corresponding mutants do not assemble the NDH5 containing complex. The second part of the paper deals with a NDH5/PnsB2 ortholog in *Physcomitrella*. The corresponding mutant is deficient in NDH activity and expression of the gene in *Arabidopsis* mutants indicates complementation of NDH5 but not PnsB2. The authors concluded from this data that the PnsB2 subunit evolved from the potential assembly factor NDH5. The work is interesting, but multiple issues raised:

Firstly, by comparing the presented work with the paper of Ishida et al. (<https://doi.org/10.1093/pcp/pcn205>), the novelty is rather low. Ishida et al. discovered NDH5 in *Arabidopsis* and showed that it is essential for accumulation of active NDH complexes. Moreover, they have identified already the NDH5 ortholog in *Physcomitrella*.

--- Ishida et al (2009) is a starting point of our work and does not affect the novelty of this manuscript. They reported that NDF5 was essential for NDH activity but did not clarify its molecular function. They found the homolog but it was unclear whether it was NDF5 or PnsB2. In this study, we clarified that NDF5 is not a subunit but an assembly factor of SubB. We also clarified a subunit (PnsB2) was evolved from an assembly factor (NDF5). The message of this manuscript is different from that in Ishida et al. 2009.

Secondly, the NDH5 containing complex was not really defined biochemically. The

authors observed comigration of selected subB components with NDH5 by sucrose density centrifugation and immunoblot analysis, but the biochemical nature of this complex remained unclear. The manuscript needs more data that show the composition of this complex (e.g. by mass spectrometry based complexome profiling, specific isolation of the complex or other techniques).

--- We tried the Co-IP of NDF5-HA but it did not work probably because of low abundance of an assembly intermediate and/or the fragile nature of the complex. Our strategy was the SDG analyses in a series of mutants and the results support the conclusion.

Thirdly, almost all data are based on single experiments or at least only data of single experiments were shown. The authors should add a thorough statistical evaluation of the results, e.g. for analysis of the NDH5 complementation mutant. How significant are the differences in the fluorescence signals? What is the variation between single measurements? The same applies in principle for the immunoblot data.

--- In the *ndf5* complementation analysis in Arabidopsis, we experienced a severe problem of *Gene X* co-suppression in the T2 generation. It was necessary to analyze individual T1 plants as showed in Fig. 5 and Supplementary Figure 11. It is scientifically sound to show all results of individual plants rather than showing the statistical analysis. On the fluorescence analysis, this information is qualitative but not very quantitative. We have used the fluorescence change for map-based cloning of at least seven genes (*crr1 – crr7*) related to the NDH complex, indicating that the phenotype could be used for the genotyping in more than 1000 T2 plants without any mistakes. Statistical analysis is not be appropriate to this result. Furthermore, the results of fluorescence analysis were confirmed by the direct protein analyses. We also showed the independent results of immunoblotting in Supplementary Fig. 11.

Fourthly, the characterization of Physcomitrella protein X is very preliminary. The Physcomitrella complementation mutant is missing and no data was presented that corroborated the function in this organism.

--- The gene was knocked out by targeting in Physcomitrella and we characterized three independent lines. This would be the standard procedure, as described in many papers

including (Kobayashi et al., 2020, *Plant Physiol.*; <https://doi.org/10.1104/pp.20.00763>). Based on the discussion with the editor, we did not perform the complementation. The antibody against Protein X in *Physcomitrella* was essential in biochemistry but was not successfully raised. We evaluated its function in *Arabidopsis ndf5* and *pnsb2* mutants. Biochemistry was performed on NDF5 in *Arabidopsis*.